# DINGO: Distributed Newton-Type Method for Gradient-Norm Optimization

**Rixon Crane**
University of Queensland
r.crane@uq.edu.au

**Fred Roosta**
University of Queensland
fred.roosta@uq.edu.au

## Abstract

For optimization of a large sum of functions in a distributed computing environment, we present a novel communication efficient Newton-type algorithm that enjoys a variety of advantages over similar existing methods. Our algorithm, DINGO, is derived by optimization of the gradient's norm as a surrogate function. DINGO does not impose any specific form on the underlying functions and its application range extends far beyond convexity and smoothness. The underlying sub-problems of DINGO are simple linear least-squares, for which a plethora of efficient algorithms exist. DINGO involves a few hyper-parameters that are easy to tune and we theoretically show that a strict reduction in the surrogate objective is guaranteed, regardless of the selected hyper-parameters.

## 1 Introduction

Consider the optimization problem

$$\min_{\mathbf{w} \in \mathbb{R}^d} \left\{ f(\mathbf{w}) \triangleq \frac{1}{m} \sum_{i=1}^m f_i(\mathbf{w}) \right\}, \tag{1}$$

in a centralized distributed computing environment involving one *driver* machine and $m$ *worker* machines, in which the $i$th worker can only locally access the $i$th component function, $f_i$. Such distributed computing settings arise increasingly more frequently as a result of technological and communication advancements that have enabled the collection of and access to large scale datasets.

As a concrete example, take a data fitting application, in which given $n$ data points, $\{\mathbf{x}_i\}_{i=1}^n$, and their corresponding loss, $\ell_i(\mathbf{w}; \mathbf{x}_i)$, parameterized by $\mathbf{w}$, the goal is to minimize the overall loss as $\min_{\mathbf{w} \in \mathbb{R}^d} \sum_{i=1}^n \ell_i(\mathbf{w}; \mathbf{x}_i)/n$. Such problems appear frequently in machine learning, e.g., [1, 2, 3] and scientific computing, e.g., [4, 5, 6]. However, in "big data" regimes where $n \gg 1$, lack of adequate computational resources, in particular storage, can severely limit, or even prevent, any attempts at solving such optimization problems in a traditional stand-alone way, e.g., using a single machine. This can be remedied through *distributed computing*, in which resources across a network of stand-alone computational nodes are "pooled" together so as to scale to the problem at hand [7]. In such a setting, where $n$ data points are distributed across $m$ workers, one can instead consider (1) with

$$f_i(\mathbf{w}) \triangleq \frac{1}{|S_i|} \sum_{j \in S_i} \ell_j(\mathbf{w}; \mathbf{x}_j), \quad i = 1, 2, \dots, m, \tag{2}$$

where $S_i \subseteq \{1, 2, \dots, n\}$, with cardinality denoted by $|S_i|$, correspond to the distribution of data across the nodes, i.e., the $i$th node has access to a portion of the data indexed by the set $S_i$.

In distributed settings, the amount of communications, i.e., messages exchanged across the network, are often considered a major bottleneck of computations (often more so than local computation

times), as they can be expensive in terms of both physical resources and time through latency [8, 9]. First-order methods [10], e.g., stochastic gradient descent (SGD) [11], solely rely on gradient information and as a result are rather easy to implement in distributed settings. They often require the performance of many computationally inexpensive iterations, which can be suitable for execution on a single machine. However, as a direct consequence, they can incur excessive communication costs in distributed environments and, hence, they might not be able to take full advantage of the available distributed computational resources.

By employing curvature information in the form of the Hessian matrix, second-order methods aim at transforming the gradient such that it is a more suitable direction to follow. Compared with first-order alternatives, although second-order methods perform more computations per iteration, they often require far fewer iterations to achieve similar results. In distributed settings, this feature can directly translate to significantly less communication costs. As a result, distributed second-order methods have the potential to become the method of choice for distributed optimization tasks.

**Notation**

We let $\langle \cdot, \cdot \rangle$ denote the common *Euclidean inner product* defined by $\langle \mathbf{x}, \mathbf{y} \rangle = \mathbf{x}^T \mathbf{y}$ for $\mathbf{x}, \mathbf{y} \in \mathbb{R}^d$. Given a vector $\mathbf{v}$ and matrix $\mathbf{A}$, we denote their vector $\ell_2$ norm and matrix *spectral* norm as $\|\mathbf{v}\|$ and $\|\mathbf{A}\|$, respectively. For $\mathbf{x}, \mathbf{z} \in \mathbb{R}^d$ we let $[\mathbf{x}, \mathbf{z}] \triangleq \{\mathbf{x} + \tau(\mathbf{z} - \mathbf{x}) \mid 0 \le \tau \le 1\}$. The *range* and *null space* of a matrix $\mathbf{A}$ is denoted by $\mathcal{R}(\mathbf{A})$ and $\mathcal{N}(\mathbf{A})$, respectively. The *Moore–Penrose inverse* [12] of $\mathbf{A}$ is denoted by $\mathbf{A}^{\dagger}$. We let $\mathbf{w}_t \in \mathbb{R}^d$ denote the point at iteration $t$. For notational convenience, we denote $\mathbf{g}_{t,i} \triangleq \nabla f_i(\mathbf{w}_t)$, $\mathbf{H}_{t,i} \triangleq \nabla^2 f_i(\mathbf{w}_t)$, $\mathbf{g}_t \triangleq \nabla f(\mathbf{w}_t)$ and $\mathbf{H}_t \triangleq \nabla^2 f(\mathbf{w}_t)$. We also let

$$\tilde{\mathbf{H}}_{t,i} \triangleq \begin{bmatrix} \mathbf{H}_{t,i} \\ \phi \mathbf{I} \end{bmatrix} \in \mathbb{R}^{2d \times d} \quad \text{and} \quad \tilde{\mathbf{g}}_t \triangleq \begin{pmatrix} \mathbf{g}_t \\ \mathbf{0} \end{pmatrix} \in \mathbb{R}^{2d}, \tag{3}$$

where $\phi > 0$, $\mathbf{I}$ is the identity matrix, and $\mathbf{0}$ is the zero vector.

**Related Work and Contributions**

Owing to the above-mentioned potential, many distributed second-order optimization algorithms have recently emerged to solve (1). Among them, most notably are GIANT [13], DiSCO [9], DANE [14], InexactDANE and AIDE [15]. While having many advantages, each of these methods respectively come with several disadvantages that can limit their applicability in certain regimes. Namely, some rely on, rather stringent, (strong) convexity assumptions, while for others the underlying sub-problems involve non-linear optimization problems that are themselves non-trivial to solve. A subtle, yet potentially severe, draw-back for many of the above-mentioned methods is that their performance can be sensitive to, and severely affected by, the choice of their corresponding hyper-parameters.

Here, we present a novel communication efficient distributed second-order optimization method that aims to alleviate many of the aforementioned disadvantages. Our approach is inspired by and follows many ideas of recent results on Newton-MR [16], which extends the application range of the classical Newton-CG beyond (strong) convexity and smoothness. More specifically, our algorithm, named DINGO for "**DI**stributed **N**ewton-type method for **G**radient-norm **O**ptimization", is derived by optimization of the gradient's norm as a surrogate function for (1), i.e.,

$$\min_{\mathbf{w} \in \mathbb{R}^d} \left\{ \frac{1}{2} \|\nabla f(\mathbf{w})\|^2 = \frac{1}{2m^2} \left\| \sum_{i=1}^{m} \nabla f_i(\mathbf{w}) \right\|^2 \right\}. \tag{4}$$

When $f$ is *invex*, [17, 18], the problems (1) and (4) have the same solutions. Recall that invexity is the generalization of convexity, which extends the sufficiency of the first order optimality condition, e.g., Karush-Kuhn-Tucker conditions, to a broader class of problems than simple convex programming. In other words, *invexity is a special case of non-convexity, which subsumes convexity as a sub-class*. In this light, unlike DiSCO and GIANT, by considering the surrogate function (4), DINGO's application range and theoretical guarantees extend far beyond convex settings to invex problems. Naturally, by considering (4), DINGO may converge to a local maximum or saddle point in non-invex problems.

Similar to GIANT and DiSCO, and in contrast to DANE, InexactDANE and AIDE, our algorithm involves a few hyper-parameters that are easy to tune and the underlying sub-problems are simple linear least-squares, for which a plethora of efficient algorithms exist. However, the theoretical

Table 1: Comparison of problem class, function form and data distribution. Note that DINGO doesn't assume invexity in analysis, rather it is suited to invex problems in practice.

|  | Problem Class | Function Form | Data Distribution |
|---|---|---|---|
| **DINGO** | Invex | Any | Any |
| **GIANT** | Strongly Convex | $\ell_j(\mathbf{w}; \mathbf{x}_j) = \psi_j(\langle \mathbf{w}, \mathbf{x}_j \rangle) + \gamma\|\mathbf{w}\|^2$ | $|S_i| > d$ |
| **DiSCO** | Strongly Convex | Any | Any |
| **InexactDANE** | Non-Convex | Any | Any |
| **AIDE** | Non-Convex | Any | Any |

Table 2: Comparison of number of sub-problem hyper-parameters and communication rounds per iteration. Under inexact update, the choice of sub-problem solver will determine additional hyper-parameters. Most communication rounds of DiSCO arise when iteratively solving its sub-problem. We assume DINGO and GIANT use two communication rounds for line-search per iteration.

|  | Number of Sub-Problem Hyper-Parameters (Under Exact Update) | Communication Rounds Per Iteration (Under Inexact Update) |
|---|---|---|
| **DINGO** | 2 | $\leq 8$ |
| **GIANT** | 0 | 6 |
| **DiSCO** | 0 | $2 + 2 \cdot (\text{sub-problem iterations})$ |
| **InexactDANE** | 2 | 4 |
| **AIDE** | 3 | $4 \cdot (\text{inner InexactDANE iterations})$ |

analysis of both GIANT and DiSCO is limited to the case where each $f_i$ is strongly convex, and for GIANT they are also of the special form where in (2) we have $\ell_j(\mathbf{w}; \mathbf{x}_j) = \psi_j(\langle \mathbf{w}, \mathbf{x}_j \rangle) + \gamma\|\mathbf{w}\|^2$, $\gamma > 0$ is a regularization parameter and $\psi_j$ is convex, e.g., linear predictor models. In contrast, DINGO does not impose any specific form on the underlying functions. Also, unlike GIANT, we allow for $|S_i| < d$ in (2). Moreover, we theoretically show that DINGO is not too sensitive to the choice of its hyper-parameters in that a strict reduction in the gradient norm is guaranteed, regardless of the selected hyper-parameters. See Tables 1 and 2 for a summary of high-level algorithm properties. Finally, we note that, unlike GIANT, DiSCO, InexactDANE and AIDE, our theoretical analysis requires exact solutions to the sub-problems. Despite the fact that the sub-problems of DINGO are simple ordinary least-squares, and that DINGO performs well in practice with very crude solutions, this is admittedly a theoretical restriction, which we aim to address in future.

The distributed computing environment that we consider is also assumed by GIANT, DiSCO, DANE, InexactDANE and AIDE. Moreover, as with these methods, we restrict communication to vectors of size linear in $d$, i.e., $\mathcal{O}(d)$. A *communication round* is performed when the driver uses a *broadcast* operation to send information to one or more workers in parallel, or uses a *reduce* operation to receive information from one or more workers in parallel. For example, computing the gradient at iteration $t$, namely $\mathbf{g}_t = \sum_{i=1}^{m} \mathbf{g}_{t,i}/m$, requires two communication rounds, i.e., the driver broadcasts $\mathbf{w}_t$ to all workers and then, by a reduce operation, receives $\mathbf{g}_{t,i}$ for all $i$. We further remind that the distributed computational model considered here is such that *the main bottleneck involves the communications across the network*.

## 2 DINGO

In this section, we describe the derivation of DINGO, as depicted in Algorithm 1. Each iteration $t$ involves the computation of two main ingredients: an *update direction* $\mathbf{p}_t$, and an appropriate *step-size* $\alpha_t$. As usual, our next iterate is then set as $\mathbf{w}_{t+1} = \mathbf{w}_t + \alpha_t \mathbf{p}_t$.

**Update Direction**

We begin iteration $t$ by distributively computing the gradient $\mathbf{g}_t$. Thereafter, we distributively compute the Hessian-gradient product $\mathbf{H}_t\mathbf{g}_t = \sum_{i=1}^m \mathbf{H}_{t,i}\mathbf{g}_t/m$ as well as the vectors $\sum_{i=1}^m \mathbf{H}_{t,i}^\dagger\mathbf{g}_t/m$ and $\sum_{i=1}^m \tilde{\mathbf{H}}_{t,i}^\dagger\tilde{\mathbf{g}}_t/m$. Computing the update direction $\mathbf{p}_t$ involves three cases, all of which involve simple linear least-squares sub-problems:

**Case 1** If $\langle\sum_{i=1}^m \mathbf{H}_{t,i}^\dagger\mathbf{g}_t/m, \mathbf{H}_t\mathbf{g}_t\rangle \geq \theta\|\mathbf{g}_t\|^2$, where $\theta$ is as in Algorithm 1, then we let $\mathbf{p}_t = \sum_{i=1}^m \mathbf{p}_{t,i}/m$, with $\mathbf{p}_{t,i} = -\mathbf{H}_{t,i}^\dagger\mathbf{g}_t$. Here, we check that the potential update direction "$-\sum_{i=1}^m \mathbf{H}_{t,i}^\dagger\mathbf{g}_t/m$" is a suitable descent direction for our surrogate objective (4). We do this since we have not imposed any restrictive assumptions on (1), e.g., strong convexity of each $f_i$, that would automatically guarantee descent; see Lemma 1 for an example of such restrictive assumptions.

**Case 2** If **Case 1** fails, we include regularization and check again that the new potential update direction yields suitable descent. Namely, if $\langle\sum_{i=1}^m \tilde{\mathbf{H}}_{t,i}^\dagger\tilde{\mathbf{g}}_t/m, \mathbf{H}_t\mathbf{g}_t\rangle \geq \theta\|\mathbf{g}_t\|^2$, then we let $\mathbf{p}_t = \sum_{i=1}^m \mathbf{p}_{t,i}/m$, with $\mathbf{p}_{t,i} = -\tilde{\mathbf{H}}_{t,i}^\dagger\tilde{\mathbf{g}}_t$.

**Case 3** If all else fails, we enforce descent in the norm of the gradient. More specifically, as **Case 2** does not hold, the set

$$\mathcal{I}_t \triangleq \left\{i = 1, 2, \ldots, m \mid \langle\tilde{\mathbf{H}}_{t,i}^\dagger\tilde{\mathbf{g}}_t, \mathbf{H}_t\mathbf{g}_t\rangle < \theta\|\mathbf{g}_t\|^2\right\}, \tag{5}$$

is non-empty. In parallel, the driver broadcasts $\mathbf{H}_t\mathbf{g}_t$ to each worker $i \in \mathcal{I}_t$ and has it locally compute the solution to

$$\arg\min_{\mathbf{p}_{t,i}} \quad \frac{1}{2}\|\mathbf{H}_{t,i}\mathbf{p}_{t,i} + \mathbf{g}_t\|^2 + \frac{\phi^2}{2}\|\mathbf{p}_{t,i}\|^2, \quad \text{such that} \quad \langle\mathbf{p}_{t,i}, \mathbf{H}_t\mathbf{g}_t\rangle \leq -\theta\|\mathbf{g}_t\|^2,$$

where $\phi$ is as in (3). It is easy to show that the solution to this problem is

$$\mathbf{p}_{t,i} = -\tilde{\mathbf{H}}_{t,i}^\dagger\tilde{\mathbf{g}}_t - \lambda_{t,i}(\tilde{\mathbf{H}}_{t,i}^T\tilde{\mathbf{H}}_{t,i})^{-1}\mathbf{H}_t\mathbf{g}_t, \quad \lambda_{t,i} = \frac{-\mathbf{g}_t^T\mathbf{H}_t\tilde{\mathbf{H}}_{t,i}^\dagger\tilde{\mathbf{g}}_t + \theta\|\mathbf{g}_t\|^2}{\mathbf{g}_t^T\mathbf{H}_t(\tilde{\mathbf{H}}_{t,i}^T\tilde{\mathbf{H}}_{t,i})^{-1}\mathbf{H}_t\mathbf{g}_t}. \tag{6}$$

The term $\lambda_{t,i}$ in (6) is positive by the definition of $\mathcal{I}_t$ and well-defined by Assumption 5, which implies that for $\mathbf{g}_t \neq \mathbf{0}$ we have $\mathbf{H}_t\mathbf{g}_t \neq \mathbf{0}$. In conclusion, for **Case 3**, each worker $i \in \mathcal{I}_t$ computes (6) and, using a reduce operation, the driver then computes the update direction $\mathbf{p}_t = \sum_{i=1}^m \mathbf{p}_{t,i}/m$, which by construction yields descent in the surrogate objective (4). Note that $\mathbf{p}_{t,i} = -\tilde{\mathbf{H}}_{t,i}^\dagger\tilde{\mathbf{g}}_t$ for all $i \notin \mathcal{I}_t$ have already been obtained as part of **Case 2**.

**Remark 1.** *The three cases help avoid the need for any unnecessary assumptions on data distribution or the knowledge of any practically unknowable constants. In fact, given Lemma 1, which imposes a certain assumption on the data distribution, we could have stated our algorithm in its simplest form, i.e., with only **Case 1**. This would be more in line with some prior works, e.g., GIANT, but it would have naturally restricted the applicability of our method in terms of data distributions.*

**Remark 2.** *In practice, like GIANT and DiSCO, our method DINGO never requires the computation or storage of an explicitly formed Hessian. Instead, it only requires Hessian-vector products, which can be computed at a similar cost to computing the gradient itself. Computing matrix pseudo-inverse and vector products, e.g., $\mathbf{H}_{t,i}^\dagger\mathbf{g}_t$, constitute the sub-problems of our algorithm. This, in turn, is done through solving least-squares problems using iterative methods that only require matrix-vector products (see Section 4 for some such methods). Thus DINGO is suitable for large dimension $d$ in (1).*

**Line Search**

After computing the update direction $\mathbf{p}_t$, DINGO computes the next iterate $\mathbf{w}_{t+1}$ by moving along $\mathbf{p}_t$ by an appropriate step-size $\alpha_t$ and forming $\mathbf{w}_{t+1} = \mathbf{w}_t + \alpha_t\mathbf{p}_t$. We use an Armijo-type line search to choose this step-size. Specifically, as we are minimizing the norm of the gradient as a surrogate function, we choose the largest $\alpha_t \in (0, 1]$ such that

$$\|\mathbf{g}_{t+1}\|^2 \leq \|\mathbf{g}_t\|^2 + 2\alpha_t\rho\langle\mathbf{p}_t, \mathbf{H}_t\mathbf{g}_t\rangle, \tag{7}$$

for some constant $\rho \in (0, 1)$. By construction of $\mathbf{p}_t$ we always have $\langle\mathbf{p}_t, \mathbf{H}_t\mathbf{g}_t\rangle \leq -\theta\|\mathbf{g}_t\|^2$. Therefore, after each iteration we are strictly decreasing the norm of the gradient, and line-search guarantees that this occurs irrespective of all hyper-parameters of DINGO, i.e., $\theta$, $\phi$ and $\rho$.

---
**Algorithm 1** DINGO
---
1: **input** initial point $\mathbf{w}_0 \in \mathbb{R}^d$, gradient tolerance $\delta \geq 0$, maximum iterations $T$, line search parameter $\rho \in (0,1)$, parameter $\theta > 0$, and regularization parameter $\phi > 0$ as in (3).
2: **for** $t = 0, 1, 2, \ldots, T-1$ **do**
3:     Distributively compute the full gradient $\mathbf{g}_t$.
4:     **if** $\|\mathbf{g}_t\| \leq \delta$ **then**
5:         **return** $\mathbf{w}_t$
6:     **else**
7:         The driver broadcasts $\mathbf{g}_t$ and, in parallel, each worker $i$ computes $\mathbf{H}_{t,i}\mathbf{g}_t$, $\mathbf{H}_{t,i}^{\dagger}\mathbf{g}_t$ and $\tilde{\mathbf{H}}_{t,i}^{\dagger}\tilde{\mathbf{g}}_t$.
8:         By a reduce operation, the driver computes $\mathbf{H}_t\mathbf{g}_t = \frac{1}{m}\sum_{i=1}^{m}\mathbf{H}_{t,i}\mathbf{g}_t$, $\frac{1}{m}\sum_{i=1}^{m}\mathbf{H}_{t,i}^{\dagger}\mathbf{g}_t$ and $\frac{1}{m}\sum_{i=1}^{m}\tilde{\mathbf{H}}_{t,i}^{\dagger}\tilde{\mathbf{g}}_t$.
9:         **if** $\left\langle \frac{1}{m}\sum_{i=1}^{m}\mathbf{H}_{t,i}^{\dagger}\mathbf{g}_t, \mathbf{H}_t\mathbf{g}_t \right\rangle \geq \theta\|\mathbf{g}_t\|^2$ **then**
10:             Let $\mathbf{p}_t = \frac{1}{m}\sum_{i=1}^{m}\mathbf{p}_{t,i}$, with $\mathbf{p}_{t,i} = -\mathbf{H}_{t,i}^{\dagger}\mathbf{g}_t$.
11:         **else if** $\left\langle \frac{1}{m}\sum_{i=1}^{m}\tilde{\mathbf{H}}_{t,i}^{\dagger}\tilde{\mathbf{g}}_t, \mathbf{H}_t\mathbf{g}_t \right\rangle \geq \theta\|\mathbf{g}_t\|^2$ **then**
12:             Let $\mathbf{p}_t = \frac{1}{m}\sum_{i=1}^{m}\mathbf{p}_{t,i}$, with $\mathbf{p}_{t,i} = -\tilde{\mathbf{H}}_{t,i}^{\dagger}\tilde{\mathbf{g}}_t$.
13:         **else**
14:             The driver computes $\mathbf{p}_{t,i} = -\tilde{\mathbf{H}}_{t,i}^{\dagger}\tilde{\mathbf{g}}_t$ for all $i$ such that $\langle \tilde{\mathbf{H}}_{t,i}^{\dagger}\tilde{\mathbf{g}}_t, \mathbf{H}_t\mathbf{g}_t \rangle \geq \theta\|\mathbf{g}_t\|^2$.
15:             The driver broadcasts $\mathbf{H}_t\mathbf{g}_t$ to each worker $i$ such that $\langle \tilde{\mathbf{H}}_{t,i}^{\dagger}\tilde{\mathbf{g}}_t, \mathbf{H}_t\mathbf{g}_t \rangle < \theta\|\mathbf{g}_t\|^2$ and, in parallel, they compute

$$\mathbf{p}_{t,i} = -\tilde{\mathbf{H}}_{t,i}^{\dagger}\tilde{\mathbf{g}}_t - \lambda_{t,i}(\tilde{\mathbf{H}}_{t,i}^{T}\tilde{\mathbf{H}}_{t,i})^{-1}\mathbf{H}_t\mathbf{g}_t, \quad \lambda_{t,i} = \frac{-\mathbf{g}_t^{T}\mathbf{H}_t\tilde{\mathbf{H}}_{t,i}^{\dagger}\tilde{\mathbf{g}}_t + \theta\|\mathbf{g}_t\|^2}{\mathbf{g}_t^{T}\mathbf{H}_t(\tilde{\mathbf{H}}_{t,i}^{T}\tilde{\mathbf{H}}_{t,i})^{-1}\mathbf{H}_t\mathbf{g}_t}.$$

16:             Using a reduce operation, the driver computes $\mathbf{p}_t = \frac{1}{m}\sum_{i=1}^{m}\mathbf{p}_{t,i}$.
17:         **end if**
18:         Choose the largest $\alpha_t \in (0,1]$ such that $\left\|\nabla f(\mathbf{w}_t + \alpha_t\mathbf{p}_t)\right\|^2 \leq \|\mathbf{g}_t\|^2 + 2\alpha_t\rho\langle\mathbf{p}_t, \mathbf{H}_t\mathbf{g}_t\rangle$.
19:         The driver computes $\mathbf{w}_{t+1} = \mathbf{w}_t + \alpha_t\mathbf{p}_t$.
20:     **end if**
21: **end for**
22: **return** $\mathbf{w}_T$.
---

## 3 Theoretical Analysis

In this section, we present convergence results for DINGO. The reader can find proofs of lemmas and theorems in the supplementary material. For notational convenience, in our analysis we have $\mathcal{C}_1 \triangleq \{t \mid \langle \sum_{i=1}^{m}\mathbf{H}_{t,i}^{\dagger}\mathbf{g}_t/m, \mathbf{H}_t\mathbf{g}_t\rangle \geq \theta\|\mathbf{g}_t\|^2\}$, $\mathcal{C}_2 \triangleq \{t \mid \langle \sum_{i=1}^{m}\tilde{\mathbf{H}}_{t,i}^{\dagger}\tilde{\mathbf{g}}_t/m, \mathbf{H}_t\mathbf{g}_t\rangle \geq \theta\|\mathbf{g}_t\|^2, \ t \notin \mathcal{C}_1\}$, and $\mathcal{C}_3 \triangleq \{t \mid t \notin (\mathcal{C}_1 \cup \mathcal{C}_2)\}$, which are sets indexing iterations $t$ that are in **Case 1**, **Case 2** and **Case 3**, respectively. The convergence analysis under these cases are treated separately in Sections 3.2, 3.3 and 3.4. The unifying result is then simply given in Corollary 1. We begin, in Section 3.1, by establishing general underlying assumptions for our analysis. The analysis of **Case 1** and **Case 3** require their own specific assumptions, which are discussed in Sections 3.2 and 3.4, respectively.

**Remark 3.** *As long as the presented assumptions are satisfied, our algorithm converges for any choice of $\theta$ and $\phi$, i.e., these hyper-parameters do not require the knowledge of the, practically unknowable, parameters from these assumptions. However, in Lemma 3 we give qualitative guidelines for a better choice of $\theta$ and $\phi$ to avoid **Case 2** and **Case 3**, which are shown to be less desirable than **Case 1**.*

### 3.1 General Assumptions

As DINGO makes use of Hessian-vector products, we make the following straightforward assumption.

**Assumption 1** (Twice Differentiability). *The functions $f_i$ in (1) are twice differentiable.*

Notice that we do not require each $f_i$ to be twice *continuously* differentiable. In particular, our analysis carries through even if the Hessian is discontinuous. This is in sharp contrast to popular belief that the application of non-smooth Hessian can hurt more so than it helps, e.g., [19]. Note that

even if the Hessian is discontinuous, Assumption 1 is sufficient in ensuring that $\mathbf{H}_{t,i}$ is symmetric, for all $t$ and $i$, [20]. Following [16], we also make the following general assumption on $f$.

**Assumption 2** (Moral-Smoothness [16]). *For all iterations $t$, there exists a constant $L \in (0, \infty)$ such that $\left\| \nabla^2 f(\mathbf{w}) \nabla f(\mathbf{w}) - \nabla^2 f(\mathbf{w}_t) \nabla f(\mathbf{w}_t) \right\| \leq L \| \mathbf{w} - \mathbf{w}_t \|$, for all $\mathbf{w} \in [\mathbf{w}_t, \mathbf{w}_t + \mathbf{p}_t]$, where $\mathbf{p}_t$ is the update direction of DINGO at iteration $t$.*

As discussed in [16] with explicit examples, Assumption 2 is *strictly weaker* than the common assumptions of the gradient and Hessian being both Lipschitz continuous. Using [16, Lemma 10], it follows from Assumptions 1 and 2 that

$$\left\| \nabla f(\mathbf{w}_t + \alpha \mathbf{p}_t) \right\|^2 \leq \left\| \mathbf{g}_t \right\|^2 + 2\alpha \langle \mathbf{p}_t, \mathbf{H}_t \mathbf{g}_t \rangle + \alpha^2 L \| \mathbf{p}_t \|^2, \tag{8}$$

for all $\alpha \in [0, 1]$ and all iterations $t$.

## 3.2 Analysis of Case 1

In this section, we analyze the convergence of iterations of DINGO that fall under **Case 1**. For such iterations, we make the following assumption about the action of the pseudo-inverse of $\mathbf{H}_{t,i}$ on $\mathbf{g}_t$.

**Assumption 3** (Pseudo-Inverse Regularity of $\mathbf{H}_{t,i}$ on $\mathbf{g}_t$). *For all $t \in \mathcal{C}_1$ and all $i = 1, 2, \ldots, m$, there exists constants $\gamma_i \in (0, \infty)$ such that $\| \mathbf{H}_{t,i}^{\dagger} \mathbf{g}_t \| \leq \gamma_i \| \mathbf{g}_t \|$.*

Assumption 3 may appear unconventional. However, it may be seen as more general than the following assumption.

**Assumption 4** (Pseudo-Inverse Regularity of $\mathbf{H}_t$ on its Range Space [16]). *There exists a constant $\gamma \in (0, \infty)$ such that for all iterates $\mathbf{w}_t$ we have $\| \mathbf{H}_t \mathbf{p} \| \geq \gamma \| \mathbf{p} \|$ for all $\mathbf{p} \in \mathcal{R}(\mathbf{H}_t)$.*

Assumption 4 implies $\| \mathbf{H}_t^{\dagger} \mathbf{g}_t \| = \| \mathbf{H}_t^{\dagger} (\mathbf{U}_t \mathbf{U}_t^T + \mathbf{U}_t^{\perp} (\mathbf{U}_t^{\perp})^T) \mathbf{g}_t \| = \| \mathbf{H}_t^{\dagger} \mathbf{U}_t \mathbf{U}_t^T \mathbf{g}_t \| \leq \gamma^{-1} \| \mathbf{g}_t \|$, where $\mathbf{U}_t$ and $\mathbf{U}_t^{\perp}$ denote arbitrary orthonormal bases for $\mathcal{R}(\mathbf{H}_t)$ and $\mathcal{R}(\mathbf{H}_t)^{\perp}$, respectively, and $\mathcal{R}(\mathbf{H}_t)^{\perp} = \mathcal{N}(\mathbf{H}_t^T) = \mathcal{N}(\mathbf{H}_t^{\dagger})$. Recall that Assumption 4 is a significant relaxation of strong convexity. As an example, an under-determined least-squares problem $f(\mathbf{w}) = \| \mathbf{A}\mathbf{w} - \mathbf{b} \|^2 / 2$, which is clearly not strongly convex, satisfies Assumption 4 with $\gamma = \sigma_{\min}^2(\mathbf{A})$, where $\sigma_{\min}(\mathbf{A})$ is the smallest non-zero singular value of $\mathbf{A}$.

**Theorem 1** (Convergence Under Case 1). *Suppose we run DINGO. Then under Assumptions 1, 2 and 3, for all $t \in \mathcal{C}_1$ we have $\| \mathbf{g}_{t+1} \|^2 \leq (1 - 2\tau_1 \rho \theta) \| \mathbf{g}_t \|^2$, where $\tau_1 = \min \left\{ 1, 2(1-\rho)\theta/(L\gamma^2) \right\}$, $\gamma = \sum_{i=1}^{m} \gamma_i / m$, $L$ is as in Assumption 2, $\gamma_i$ are as in Assumption 3, $\rho$ and $\theta$ are as in Algorithm 1.*

From the proof of Theorem 1, it is easy to see that $\forall t \in \mathcal{C}_1$ we are guaranteed that $0 < 1 - 2\tau_1 \rho \theta < 1$. In Theorem 1, the term $\gamma$ is the *average* of the $\gamma_i$'s. This is beneficial as it "smooths out" non-uniformity in $\gamma_i$'s; for example, $\gamma \geq \min_i \gamma_i$. Under specific assumptions on (1), we can theoretically guarantee that $t \in \mathcal{C}_1$ for all iterations $t$. The following lemma provides one such example.

**Lemma 1.** *Suppose Assumption 1 holds and that we run DINGO. Furthermore, suppose that for all iterations $t$ and all $i = 1, 2, \ldots, m$, the Hessian matrix $\mathbf{H}_{t,i}$ is invertible and there exists constants $\varepsilon_i \in [0, \infty)$ and $\nu_i \in (0, \infty)$ such that $\| \mathbf{H}_{t,i} - \mathbf{H}_t \| \leq \varepsilon_i$ and $\nu_i \| \mathbf{g}_t \| \leq \| \mathbf{H}_{t,i} \mathbf{g}_t \|$. If $\sum_{i=1}^{m} (1 - \varepsilon_i/\nu_i)/m \geq \theta$ then $t \in \mathcal{C}_1$ for all $t$, where $\theta$ is as in Algorithm 1.*

As an example, the Assumptions of Lemma 1 trivially hold if each $f_i$ is strongly convex and we assume certain data distribution. Under the assumptions of Lemma 1, if the Hessian matrix for each worker is *on average* a reasonable approximation to the full Hessian, i.e., $\varepsilon_i$ is on average sufficiently small so that $\sum_{i=1}^{m} \varepsilon_i/\nu_i < m$, then we can choose $\theta$ small enough to ensure that $t \in \mathcal{C}_1$ for all $t$. In other words, for the iterates to stay in $\mathcal{C}_1$, we do not require the Hessian matrix of each individual worker to be a high-quality approximation to full Hessian (which could indeed be hard to enforce in many practical applications). As long as the data is distributed in such a way that Hessian matrices are on average reasonable approximations, we can guarantee to have $t \in \mathcal{C}_1$ for all $t$.

## 3.3 Analysis of Case 2

We now analyze the convergence of DINGO for iterations that fall under **Case 2**. For this case, we do not require any additional assumptions to that of Assumptions 1 and 2. Instead, we use the upper

bound: $\|\tilde{\mathbf{H}}_{t,i}^\dagger\| \leq 1/\phi$ for all iterations $t$ and all $i = 1, 2, \ldots, m$, where $\phi$ is as in Algorithm 1; see Lemma 4 in the supplementary material for a proof of this upper bound.

**Theorem 2** (Convergence Under Case 2). *Suppose we run DINGO. Then under Assumptions 1 and 2, for all $t \in \mathcal{C}_2$ we have $\|\mathbf{g}_{t+1}\|^2 \leq (1 - 2\tau_2\rho\theta)\|\mathbf{g}_t\|^2$, where $\tau_2 = \min\left\{1, 2(1-\rho)\phi^2\theta/L\right\}$, $L$ is as in Assumption 2, and $\rho, \theta$ and $\phi$ are as in Algorithm 1.*

In our experience, we have found that **Case 2** does not occur frequently in practice. It serves more of a theoretical purpose and is used to identify when **Case 3** is required. **Case 2** may be thought of as a specific instance of **Case 3**, in which $\mathcal{I}_t$ is empty. However, it merits its own case, as in analysis it does not require additional assumptions to Assumptions 1 and 2, and in practice it may avoid an additional two communication rounds. If we were to bypass **Case 2** to **Case 3** and allow $\mathcal{I}_t$ to be empty, then Theorem 3 of Section 3.4 with $|\mathcal{I}_t| = 0$, which states the convergence for **Case 3**, indeed coincides with Theorem 2.

### 3.4 Analysis of Case 3

Now we turn to the final case, and analyze the convergence of iterations of DINGO that fall under **Case 3**. For such iterations, we make the following assumption.

**Assumption 5.** *For all $t \in \mathcal{C}_3$ and all $i = 1, 2, \ldots, m$ there exists constants $\delta_i \in (0, \infty)$ such that $\left\|(\tilde{\mathbf{H}}_{t,i}^T)^\dagger \mathbf{H}_t \mathbf{g}_t\right\| \geq \delta_i \|\mathbf{g}_t\|$.*

Assumption 5, like Assumption 3, may appear unconventional. In Lemma 2 we show how Assumption 5 is implied by three other reasonable assumptions, one of which is as follows.

**Assumption 6** (Gradient-Hessian Null-Space Property [16]). *There exists a constant $\nu \in (0, 1]$ such that $\left\|(\mathbf{U}_\mathbf{w}^\perp)^T \nabla f(\mathbf{w})\right\|^2 \leq (1 - \nu)\nu^{-1}\left\|\mathbf{U}_\mathbf{w}^T \nabla f(\mathbf{w})\right\|^2$, for all $\mathbf{w} \in \mathbb{R}^d$, where $\mathbf{U}_\mathbf{w}$ and $\mathbf{U}_\mathbf{w}^\perp$ denote any orthonormal bases for $\mathcal{R}\left(\nabla^2 f(\mathbf{w})\right)$ and its orthogonal complement, respectively.*

Assumption 6 implies that, as the iterations progress, the gradient will not become arbitrarily orthogonal to the range space of the Hessian matrix. As an example, any least-squares problem $f(\mathbf{w}) = \|\mathbf{A}\mathbf{w} - \mathbf{b}\|^2/2$ satisfies Assumption 6 with $\nu = 1$; see [16] for detailed discussion and many more examples of Assumption 6.

**Lemma 2.** *Suppose Assumptions 4 and 6 hold and $\|\mathbf{H}_{t,i}\|^2 \leq \tau_i$, $\forall t \in \mathcal{C}_3$, $i = 1, 2, \ldots, m$, $\tau_i \in (0, \infty)$, i.e., local Hessians are bounded. Then, Assumption 5 holds with $\delta_i = \gamma\sqrt{\nu/(\tau_i + \phi^2)}$, where $\phi$ is as in Algorithm 1, and $\gamma$ and $\nu$ are as in Assumptions 4 and 6, respectively.*

The following theorem provides convergence properties for iterations of DINGO that are in **Case 3**.

**Theorem 3** (Convergence Under Case 3). *Suppose we run DINGO. Then under Assumptions 1, 2 and 5, for all $t \in \mathcal{C}_3$ we have $\|\mathbf{g}_{t+1}\|^2 \leq (1 - 2\omega_t\rho\theta)\|\mathbf{g}_t\|^2 \leq (1 - 2\tau_3\rho\theta)\|\mathbf{g}_t\|^2$, where $\omega_t = \min\{1, 2(1-\rho)\theta/Lc_t^2\}$, $\tau_3 = \min\{1, 2(1-\rho)\theta/Lc^2\}$,*

$$c_t = \frac{1}{m\phi}\left(m + |\mathcal{I}_t| + \theta\sum_{i \in \mathcal{I}_t}\frac{1}{\delta_i}\right), \quad c = \frac{2}{\phi} + \frac{\theta}{m\phi}\sum_{i=1}^m \frac{1}{\delta_i},$$

*$L$ is as in Assumption 2, $\delta_i$ are as in Assumption 5, $\mathcal{I}_t$ is as in (5), and $\rho, \theta$ and $\phi$ are as in Algorithm 1.*

Note that the convergence in Theorem 3 is given in both iteration dependent and independent format, since the former explicitly relates the convergence rate to the size of $\mathcal{I}_t$, while the latter simply upper-bounds this, and hence is qualitatively less informative.

Comparing Theorems 2 and 3, iterations of DINGO should have slower convergence if they are in **Case 3** rather than **Case 2**. By Theorem 3, if an iteration $t$ resorts to **Case 3** then we may have slower convergence for larger $|\mathcal{I}_t|$. Moreover, this iteration would require two more communication rounds than if it were to stop in **Case 1** or **Case 2**. Therefore, one may wish to choose $\theta$ and $\phi$ appropriately to reduce the chances that iteration $t$ falls in **Case 3** or that $|\mathcal{I}_t|$ is large. Under this consideration, Lemma 3 presents a necessary condition on a relationship between $\theta$ and $\phi$.

**Lemma 3.** *Suppose we run DINGO. Under Assumption 1, if $|\mathcal{I}_t| < m$ for some iteration $t$, then $\theta\phi \leq \|\mathbf{H}_t\mathbf{g}_t\|/\|\mathbf{g}_t\|$.*

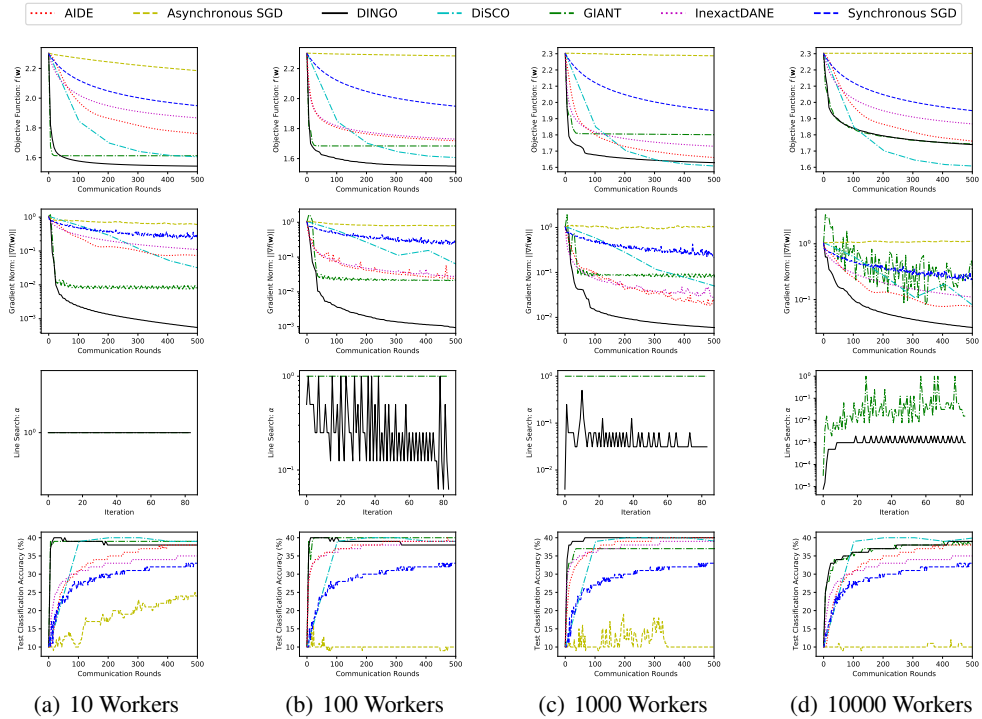

Figure 1: Softmax regression problem on the CIFAR10 dataset. All algorithms are initialized at $\mathbf{w}_0 = \mathbf{0}$. In all plots, Sync-SGD has a learning rate of $10^{-2}$. Async-SGD has a learning rate of: $10^{-3}$ in 1(a), $10^{-4}$ in 1(b) and 1(c), and $10^{-5}$ in 1(d). SVRG has a learning rate of: $10^{-3}$ in 1(a) and 1(d), and $10^{-2}$ in 1(b) and 1(c). AIDE has $\tau = 100$ in 1(a) and 1(d), $\tau = 1$ in 1(b), and $\tau = 10$ in 1(c). The number of workers is the value of $m$ in (1) and (2).

Lemma 3 suggests that we should pick $\theta$ and $\phi$ so that their product, $\theta\phi$, is small. Clearly, choosing smaller $\theta$ will increase the chance of an iteration of DINGO being in **Case 1** or **Case 2**. However, this also gives a lower rate of convergence in Theorems 1 and 2. Choosing smaller $\phi$ will preserve more curvature information of the Hessian $\mathbf{H}_{t,i}$ in $\tilde{\mathbf{H}}_{t,i}^{\dagger}$. However, $\phi$ should still be reasonably large, as making $\phi$ smaller also makes some of the sub-problems of DINGO more ill-conditioned. There is a non-trivial trade-off between $\phi$ and $\theta$, and Lemma 3 gives an appropriate way to set them.

We can finally present a unifying result on the overall worst-case *linear convergence rate* of DINGO.

**Corollary 1** (Overall Linear Convergence of DINGO)**.** *Suppose we run DINGO. Then under Assumptions 1, 2, 3 and 5, for all iterations $t$ we have $\|\mathbf{g}_{t+1}\|^2 \leq (1 - 2\tau\rho\theta)\|\mathbf{g}_t\|^2$ with $\tau = \min\{\tau_1, \tau_2, \tau_3\}$, where $\tau_1$, $\tau_2$ and $\tau_3$ are as in Theorems 1, 2, and 3, respectively, and $\rho$ and $\theta$ are as in Algorithm 1.*

From Corollary 1, DINGO can achieve $\|\mathbf{g}_t\| \leq \varepsilon$ with $\mathcal{O}(\log(\varepsilon)/(\tau\rho\theta))$ communication rounds. Moreover, the term $\tau$ is a lower bound on the step-size under all cases, which can determine the maximum communication cost needed during line-search. For example, knowing $\tau$ could determine the number of step-sizes used in backtracking line-search for DINGO in Section 4.

## 4 Experiments

In this section, we evaluate the empirical performance of DINGO, GIANT, DiSCO, InexactDANE, AIDE, Asynchronous SGD (Async-SGD) and Synchronous SGD (Sync-SGD) [11] on the strongly convex problem of softmax cross-entropy minimization with regularization on the CIFAR10 dataset [21], see Figure 1. This dataset has 50000 training samples, 10000 test samples and each datapoint $\mathbf{x}_i \in \mathbb{R}^{3072}$ has a label $y_i \in \{1, 2, \ldots, 10\}$. This problem has dimension $d = 27648$. In the supplementary material, the reader can find additional experiments on another softmax regression

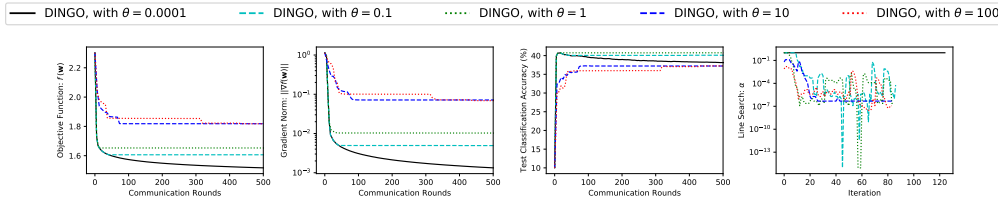

Figure 2: Softmax regression problem on the CIFAR10 dataset. We compare DINGO with $\theta = 10^{-4}, 10^{-1}, 1, 10, 100$. All iterations are in **Case 1** with $\theta = 10^{-4}$, which implies the same plot would occur for all $\theta \leq 10^{-4}$. **Case 1** and **Case 3** iterations occur when $\theta = 10^{-1}, 1$. All iterations under $\theta = 10, 100$ are in **Case 3**.

as well as on a *Gaussian mixture model* and *autoencoder* problem. In all experiments we consider (1) with (2), where the sets $S_1, S_2, \ldots, S_m$ randomly partition the index set $\{1, 2, \ldots, n\}$, with each having equal size $s = n/m$. Code is available at https://github.com/RixonC/DINGO.

We describe some implementation details. All sub-problem solvers are limited to 50 iterations and do not employ preconditioning. For DINGO, we use the sub-problem solvers MINRES-QLP [22], LSMR [23] and CG [24] when computing $\mathbf{H}_{t,i}^{\dagger}\mathbf{g}_t$, $\tilde{\mathbf{H}}_{t,i}^{\dagger}\tilde{\mathbf{g}}_t$ and $(\tilde{\mathbf{H}}_{t,i}^T \tilde{\mathbf{H}}_{t,i})^{-1}(\mathbf{H}_t\mathbf{g}_t)$, respectively. We choose CG for the latter problem as the approximation $\mathbf{x}$ of $(\tilde{\mathbf{H}}_{t,i}^T \tilde{\mathbf{H}}_{t,i})^{-1}\mathbf{H}_t\mathbf{g}_t$ is guaranteed to satisfy $\langle \mathbf{H}_t\mathbf{g}_t, \mathbf{x} \rangle > 0$ regardless of the number of CG iterations performed. For DINGO, unless otherwise stated, we set $\theta = 10^{-4}$ and $\phi = 10^{-6}$. We use backtracking line search for DINGO and GIANT to select the largest step-size in $\{1, 2^{-1}, 2^{-2}, \ldots, 2^{-50}\}$ which passes, with an Armijo line-search parameter of $10^{-4}$. For InexactDANE, we set $\eta = 1$ and $\mu = 0$, as in [15], and use SVRG [25] as a local solver with the best learning rate from $\{10^{-6}, 10^{-5}, \ldots, 10^6\}$. We have each iteration of AIDE invoke one iteration of InexactDANE, with the same parameters as in the stand-alone InexactDANE method, and use the best catalyst acceleration parameter $\tau \in \{10^{-6}, 10^{-5}, \ldots, 10^6\}$, as in [15]. For Async-SGD and Sync-SGD we report the best learning rate from $\{10^{-6}, 10^{-5}, \ldots, 10^6\}$ and each worker uses a mini-batch of size $n/(5m)$.

DiSCO has consistent performance, regardless of the number of workers, due to the distributed PCG algorithm. This essentially allows DiSCO to perform Newton's method over the full dataset. This is unnecessarily costly, in terms of communication rounds, when $s$ is reasonably large. Thus we see it perform comparatively poorly in Plots 1(a), 1(b), and 1(c). DiSCO outperforms GIANT and DINGO in Plot 1(d). This is likely because the local directions ($-\mathbf{H}_{t,i}^{-1}\mathbf{g}_t$ and $\mathbf{p}_{t,i}$ for GIANT and DINGO, respectively) give poor updates as they are calculated using very small subsets of data, i.e., in Plot 1(d) each worker has access to only 5 data points, while $d = 27648$.

A significant advantage of DINGO to InexactDANE, AIDE, Async-SGD and Sync-SGD is that it is relatively easy to tune hyper-parameters. Namely, making bad choices for $\rho$, $\theta$ and $\phi$ in DINGO will give sub-optimal performance; however, it is still theoretically guaranteed to strictly decrease the norm of the gradient. In contrast, some choices of hyper-parameters in InexactDANE, AIDE, Async-SGD and Sync-SGD will cause divergence and these choices can be problem specific. Moreover, these methods can be very sensitive to the chosen hyper-parameters with some being very difficult to select. For example, the acceleration parameter $\tau$ in AIDE was found to be difficult and time consuming to tune and the performance of AIDE was sensitive to it; notice the variation in selected $\tau$ in Figure 1. This difficulty was also observed in [13, 15]. We found that simply choosing $\rho$, $\theta$ and $\phi$ to be small, in DINGO, gave high performance. Figure 2 compares different values of $\theta$.

## 5  Future Work

The following is left for future work. First, extending the analysis of DINGO to include convergence results under inexact update. Second, finding more efficient methods of line search, for practical implementations of DINGO, than backtracking line search. Using backtracking line search for GIANT and DINGO requires the communication of some constant number of scalars and vectors, respectively. Hence, for DINGO, it may transmit a large amount of data over the network, while still only requiring two communication rounds per iteration of DINGO. Lastly, considering modifications to DINGO that prevent convergence to a local maximum/saddle point in non-invex problems.

**Acknowledgments**

Both authors gratefully acknowledge the generous support by the Australian Research Council (ARC) Centre of Excellence for Mathematical & Statistical Frontiers (ACEMS). Fred Roosta was partially supported by DARPA as well as ARC through a Discovery Early Career Researcher Award (DE180100923). Part of this work was done while Fred Roosta was visiting the Simons Institute for the Theory of Computing.

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
