[Supplementary Material]

# A  Proofs of Results From Section 3

In this section, we provide proofs of our results from Section 3.

## A.1  Theorem 1

*Proof of Theorem 1.* Let $t \in \mathcal{C}_1$ as defined in Section 3. For iteration $t$ we use the update direction $\mathbf{p}_t = \sum_{i=1}^{m} \mathbf{p}_{t,i}/m$, with $\mathbf{p}_{t,i} = -\mathbf{H}_{t,i}^{\dagger} \mathbf{g}_t$. Using Assumption 3 we obtain the following upper bound on $\|\mathbf{p}_t\|$:

$$\|\mathbf{p}_t\| = \frac{1}{m}\left\|\sum_{i=1}^{m} \mathbf{p}_{t,i}\right\| \leq \frac{1}{m}\sum_{i=1}^{m}\|\mathbf{p}_{t,i}\| \leq \gamma\|\mathbf{g}_t\|,$$

where $\gamma = \sum_{i=1}^{m} \gamma_i/m$. This and (8) imply

$$\left\|\nabla f(\mathbf{w}_t + \alpha\mathbf{p}_t)\right\|^2 \leq \|\mathbf{g}_t\|^2 + 2\alpha\langle\mathbf{p}_t, \mathbf{H}_t\mathbf{g}_t\rangle + \alpha^2\gamma^2 L\|\mathbf{g}_t\|^2, \tag{9}$$

for all $\alpha \in [0,1]$. We now fix $\alpha > 0$ such that $\alpha \leq \tau_1$, where

$$\tau_1 = \min\left\{1, \frac{2(1-\rho)\theta}{L\gamma^2}\right\}.$$

For such $\alpha$ we have

$$\alpha^2\gamma^2 L\|\mathbf{g}_t\|^2 \leq 2\alpha(1-\rho)\theta\|\mathbf{g}_t\|^2,$$

and as $\langle\mathbf{p}_t, \mathbf{H}_t\mathbf{g}_t\rangle \leq -\theta\|\mathbf{g}_t\|^2$ we obtain

$$\alpha^2\gamma^2 L\|\mathbf{g}_t\|^2 \leq 2\alpha(\rho-1)\langle\mathbf{p}_t, \mathbf{H}_t\mathbf{g}_t\rangle.$$

This implies

$$\|\mathbf{g}_t\|^2 + 2\alpha\langle\mathbf{p}_t, \mathbf{H}_t\mathbf{g}_t\rangle + \alpha^2\gamma^2 L\|\mathbf{g}_t\|^2 \leq \|\mathbf{g}_t\|^2 + 2\alpha\rho\langle\mathbf{p}_t, \mathbf{H}_t\mathbf{g}_t\rangle,$$

and by (9) we have

$$\left\|\nabla f(\mathbf{w}_t + \alpha\mathbf{p}_t)\right\|^2 \leq \|\mathbf{g}_t\|^2 + 2\alpha\rho\langle\mathbf{p}_t, \mathbf{H}_t\mathbf{g}_t\rangle.$$

Therefore, line search (7) will pass for some $\alpha_t \geq \tau_1$ and $\|\mathbf{g}_{t+1}\|^2 \leq (1 - 2\tau_1\rho\theta)\|\mathbf{g}_t\|^2$. □

## A.2  Lemma 1

*Proof of Lemma 1.* We have

$$\|\mathbf{g}_t\|^2 - \mathbf{g}_t^T\mathbf{H}_{t,i}^{-1}\mathbf{H}_t\mathbf{g}_t = \mathbf{g}_t^T\mathbf{H}_{t,i}^{-1}(\mathbf{H}_{t,i} - \mathbf{H}_t)\mathbf{g}_t \leq \varepsilon_i\|\mathbf{g}_t\|\|\mathbf{H}_{t,i}^{-1}\mathbf{g}_t\| \leq \frac{\varepsilon_i}{\nu_i}\|\mathbf{g}_t\|^2, \tag{10}$$

for all iterations $t$ and all $i = 1, 2, \ldots, m$. Therefore, if $\sum_{i=1}^{m}(1 - \varepsilon_i/\nu_i)/m \geq \theta$ then (10) implies

$$\left\langle\frac{1}{m}\sum_{i=1}^{m}\mathbf{H}_{t,i}^{-1}\mathbf{g}_t, \mathbf{H}_t\mathbf{g}_t\right\rangle = \frac{1}{m}\sum_{i=1}^{m}\mathbf{g}_t^T\mathbf{H}_{t,i}^{-1}\mathbf{H}_t\mathbf{g} \geq \frac{1}{m}\sum_{i=1}^{m}\left(1 - \frac{\varepsilon_i}{\nu_i}\right)\|\mathbf{g}_t\|^2 \geq \theta\|\mathbf{g}_t\|^2,$$

for all $t$, which implies $t \in \mathcal{C}_1$ for all $t$. □

## A.3  Theorem 2

**Lemma 4.** *Let $\mathbf{A}_1 \in \mathbb{R}^{m \times n}$ and $\mathbf{A}_2 \in \mathbb{R}^{n \times n}$ for arbitrary positive integers $m$ and $n$. Suppose that $\mathbf{A}_2$ is non-singular. If we let*

$$\mathbf{A} = \begin{bmatrix}\mathbf{A}_1 \\ \mathbf{A}_2\end{bmatrix} \in \mathbb{R}^{(m+n) \times n},$$

*then $\|\mathbf{A}^{\dagger}\| \leq \|\mathbf{A}_2^{-1}\|$.*

*Proof.* Recall that
$$\|\mathbf{A}^\dagger\| = \max_{\mathbf{x} \neq \mathbf{0}} \frac{\|\mathbf{A}^\dagger \mathbf{x}\|}{\|\mathbf{x}\|}.$$
For any such $\mathbf{x}$ we may write $\mathbf{x} = \mathbf{y} + \mathbf{z}$ with $\mathbf{y} \in \mathcal{R}(\mathbf{A})$ and $\mathbf{z} \in \mathcal{R}(\mathbf{A})^\perp = \mathcal{N}(\mathbf{A}^T) = \mathcal{N}(\mathbf{A}^\dagger)$. Therefore, $\mathbf{A}^\dagger \mathbf{x} = \mathbf{A}^\dagger \mathbf{y}$ and $\|\mathbf{x}\|^2 = \|\mathbf{y}\|^2 + \|\mathbf{z}\|^2$. This implies
$$\frac{\|\mathbf{A}^\dagger \mathbf{x}\|}{\|\mathbf{x}\|} = \frac{\|\mathbf{A}^\dagger \mathbf{y}\|}{\sqrt{\|\mathbf{y}\|^2 + \|\mathbf{z}\|^2}},$$
which is maximized when $\mathbf{z} = \mathbf{0}$. Thus,
$$\|\mathbf{A}^\dagger\| = \max_{\mathbf{y} \in \mathcal{R}(\mathbf{A}) \setminus \{\mathbf{0}\}} \frac{\|\mathbf{A}^\dagger \mathbf{y}\|}{\|\mathbf{y}\|}.$$
As $\mathbf{A}_2$ has full column rank then so does $\mathbf{A}$. Thus, $\mathbf{A}^\dagger$ is the left inverse of $\mathbf{A}$ and $\mathbf{A}\mathbf{v} = \mathbf{0}$ only when $\mathbf{v} = \mathbf{0}$. Therefore,
$$\|\mathbf{A}^\dagger\| = \max_{\mathbf{v} \neq \mathbf{0}} \frac{\|\mathbf{A}^\dagger \mathbf{A}\mathbf{v}\|}{\|\mathbf{A}\mathbf{v}\|} = \left( \min_{\|\mathbf{v}\|=1} \|\mathbf{A}\mathbf{v}\| \right)^{-1}.$$
This implies
$$\|\mathbf{A}^\dagger\|^{-1} = \min_{\|\mathbf{v}\|=1} \left\| \begin{bmatrix} \mathbf{A}_1 \mathbf{v} \\ \mathbf{A}_2 \mathbf{v} \end{bmatrix} \right\| \geq \min_{\|\mathbf{v}\|=1} \|\mathbf{A}_2 \mathbf{v}\| = \|\mathbf{A}_2^{-1}\|^{-1}. \quad \square$$

The following inequality follows immediately from Lemma 4:
$$\|\tilde{\mathbf{H}}_{t,i}^\dagger\| \leq \frac{1}{\phi}. \tag{11}$$
With this, we now give a proof of Theorem 2.

*Proof of Theorem 2.* Let $t \in \mathcal{C}_2$ as defined in Section 3. For iteration $t$ we use the update direction $\mathbf{p}_t = \sum_{i=1}^m \mathbf{p}_{t,i}/m$, with $\mathbf{p}_{t,i} = -\tilde{\mathbf{H}}_{t,i}^\dagger \tilde{\mathbf{g}}_t$. It follows from (11) that
$$\|\mathbf{p}_t\| \leq \frac{1}{m} \sum_{i=1}^m \|\mathbf{p}_{t,i}\| \leq \frac{1}{m} \sum_{i=1}^m \frac{1}{\phi} \|\tilde{\mathbf{g}}_t\| = \frac{1}{\phi} \|\mathbf{g}_t\|,$$
where $\phi$ is as in Algorithm 1. From this and an analogous argument to that in the proof of Theorem 1, for $\alpha \in (0, \tau_2]$, where
$$\tau_2 = \min \left\{ 1, \frac{2(1-\rho)\phi^2 \theta}{L} \right\},$$
we have
$$\left\| \nabla f(\mathbf{w}_t + \alpha \mathbf{p}_t) \right\|^2 \leq \|\mathbf{g}_t\|^2 + 2\alpha\rho \langle \mathbf{p}_t, \mathbf{H}_t \mathbf{g}_t \rangle.$$
Therefore, line search (7) will pass for some $\alpha_t \geq \tau_2$ and $\|\mathbf{g}_{t+1}\|^2 \leq (1 - 2\tau_2 \rho \theta)\|\mathbf{g}_t\|^2$. $\quad \square$

## A.4 Lemma 2

*Proof of Lemma 2.* Let $t \in \mathcal{C}_3$, as defined in Section 3, and $i = 1, 2, \ldots, m$ be arbitrary. The positive definite matrix $\tilde{\mathbf{H}}_{t,i}^T \tilde{\mathbf{H}}_{t,i} = \mathbf{H}_{t,i}^2 + \phi^2 \mathbf{I}$ has eigenvalues at most $\tau_i + \phi^2$. Hence, the matrix $(\tilde{\mathbf{H}}_{t,i}^T \tilde{\mathbf{H}}_{t,i})^{-1}$ has eigenvalues at least $(\tau_i + \phi^2)^{-1}$. Therefore,
$$\left\| (\tilde{\mathbf{H}}_{t,i}^T)^\dagger \mathbf{H}_t \mathbf{g}_t \right\|^2 = \mathbf{g}_t^T \mathbf{H}_t (\tilde{\mathbf{H}}_{t,i}^T \tilde{\mathbf{H}}_{t,i})^{-1} \mathbf{H}_t \mathbf{g}_t \geq \frac{1}{\tau_i + \phi^2} \|\mathbf{H}_t \mathbf{g}_t\|^2.$$
We also have
$$\|\mathbf{H}_t \mathbf{g}_t\| = \|\mathbf{H}_t \mathbf{U}_{\mathbf{w}_t} \mathbf{U}_{\mathbf{w}_t}^T \mathbf{g}_t\| \geq \gamma \|\mathbf{U}_{\mathbf{w}_t} \mathbf{U}_{\mathbf{w}_t}^T \mathbf{g}_t\| \geq \gamma \sqrt{\nu} \|\mathbf{g}_t\|,$$
where the first and second inequality follow from Assumptions 4 and 6, respectively, and $\mathbf{U}_{\mathbf{w}_t}$ and $\mathbf{U}_{\mathbf{w}_t}^\perp$ are as in Assumption 6. In conclusion,
$$\left\| (\tilde{\mathbf{H}}_{t,i}^T)^\dagger \mathbf{H}_t \mathbf{g}_t \right\| \geq \gamma \left( \frac{\nu}{\tau_i + \phi^2} \right)^{\frac{1}{2}} \|\mathbf{g}_t\|,$$
and this holds for all $t \in \mathcal{C}_3$ and all $i = 1, 2, \ldots, m$. $\quad \square$

## A.5 Theorem 3

*Proof of Theorem 3.* Let $t \in \mathcal{C}_3$, as defined in Section 3. The set $\mathcal{I}_t$, as defined in (5), is non-empty. Each worker $i$, for $i \in \mathcal{I}_t$, computes

$$\mathbf{p}_{t,i} = -\tilde{\mathbf{H}}_{t,i}^{\dagger}\tilde{\mathbf{g}}_t - \lambda_{t,i}(\tilde{\mathbf{H}}_{t,i}^T\tilde{\mathbf{H}}_{t,i})^{-1}\mathbf{H}_t\mathbf{g}_t,$$

$$\lambda_{t,i} = \frac{-\mathbf{g}_t^T\mathbf{H}_t\tilde{\mathbf{H}}_{t,i}^{\dagger}\tilde{\mathbf{g}}_t + \theta\|\mathbf{g}_t\|^2}{\mathbf{g}_t^T\mathbf{H}_t(\tilde{\mathbf{H}}_{t,i}^T\tilde{\mathbf{H}}_{t,i})^{-1}\mathbf{H}_t\mathbf{g}_t}.$$

The term $\lambda_{t,i}$ is both well-defined and positive by Assumption 5, lines 4 and 5 of Algorithm 1, and the definition of $\mathcal{I}_t$. It follows from $\mathbf{g}_t^T\mathbf{H}_t(\tilde{\mathbf{H}}_{t,i}^T\tilde{\mathbf{H}}_{t,i})^{-1}\mathbf{H}_t\mathbf{g}_t = \left\|(\tilde{\mathbf{H}}_{t,i}^T)^{\dagger}\mathbf{H}_t\mathbf{g}_t\right\|^2$ and inequality (11) that for all $i \in \mathcal{I}_t$ we have

$$\lambda_{t,i}\left\|(\tilde{\mathbf{H}}_{t,i}^T\tilde{\mathbf{H}}_{t,i})^{\dagger}\mathbf{H}_t\mathbf{g}_t\right\| = \left(-\mathbf{g}_t^T\mathbf{H}_t\tilde{\mathbf{H}}_{t,i}^{\dagger}\tilde{\mathbf{g}}_t + \theta\|\mathbf{g}_t\|^2\right)\frac{\left\|(\tilde{\mathbf{H}}_{t,i}^T\tilde{\mathbf{H}}_{t,i})^{-1}\mathbf{H}_t\mathbf{g}_t\right\|}{\left\|(\tilde{\mathbf{H}}_{t,i}^T)^{\dagger}\mathbf{H}_t\mathbf{g}_t\right\|^2}$$

$$= \frac{-\mathbf{g}_t^T\mathbf{H}_t\tilde{\mathbf{H}}_{t,i}^{\dagger}\tilde{\mathbf{g}}_t + \theta\|\mathbf{g}_t\|^2}{\left\|(\tilde{\mathbf{H}}_{t,i}^T)^{\dagger}\mathbf{H}_t\mathbf{g}_t\right\|} \cdot \frac{\left\|\tilde{\mathbf{H}}_{t,i}^{\dagger}\left((\tilde{\mathbf{H}}_{t,i}^T)^{\dagger}\mathbf{H}_t\mathbf{g}_t\right)\right\|}{\left\|(\tilde{\mathbf{H}}_{t,i}^T)^{\dagger}\mathbf{H}_t\mathbf{g}_t\right\|}$$

$$\leq \frac{1}{\phi}\left(\frac{-\tilde{\mathbf{g}}_t^T(\tilde{\mathbf{H}}_{t,i}^T)^{\dagger}\mathbf{H}_t\mathbf{g}_t}{\left\|(\tilde{\mathbf{H}}_{t,i}^T)^{\dagger}\mathbf{H}_t\mathbf{g}_t\right\|} + \frac{\theta\|\mathbf{g}_t\|^2}{\left\|(\tilde{\mathbf{H}}_{t,i}^T)^{\dagger}\mathbf{H}_t\mathbf{g}_t\right\|}\right).$$

Moreover, by Assumption 5, for all $i \in \mathcal{I}_t$ we have

$$\lambda_{t,i}\left\|(\tilde{\mathbf{H}}_{t,i}^T\tilde{\mathbf{H}}_{t,i})^{\dagger}\mathbf{H}_t\mathbf{g}_t\right\| \leq \frac{1}{\phi}\left(\frac{\left|\tilde{\mathbf{g}}_t^T(\tilde{\mathbf{H}}_{t,i}^T)^{\dagger}\mathbf{H}_t\mathbf{g}_t\right|}{\left\|(\tilde{\mathbf{H}}_{t,i}^T)^{\dagger}\mathbf{H}_t\mathbf{g}_t\right\|} + \frac{\theta}{\delta_i}\|\mathbf{g}_t\|\right)$$

$$\leq \frac{1}{\phi}\left(\frac{\|\tilde{\mathbf{g}}_t\| \cdot \left\|(\tilde{\mathbf{H}}_{t,i}^T)^{\dagger}\mathbf{H}_t\mathbf{g}_t\right\|}{\left\|(\tilde{\mathbf{H}}_{t,i}^T)^{\dagger}\mathbf{H}_t\mathbf{g}_t\right\|} + \frac{\theta}{\delta_i}\|\mathbf{g}_t\|\right)$$

$$= \frac{1}{\phi}\left(1 + \frac{\theta}{\delta_i}\right)\|\mathbf{g}_t\|.$$

Therefore, for all $i \in \mathcal{I}_t$ we have

$$\|\mathbf{p}_{t,i}\| \leq \|\tilde{\mathbf{H}}_{t,i}^{\dagger}\tilde{\mathbf{g}}_t\| + \lambda_{t,i}\left\|(\tilde{\mathbf{H}}_{t,i}^T\tilde{\mathbf{H}}_{t,i})^{\dagger}\mathbf{H}_t\mathbf{g}_t\right\| \leq \frac{1}{\phi}\left(2 + \frac{\theta}{\delta_i}\right)\|\mathbf{g}_t\|.$$

This implies

$$\|\mathbf{p}_t\| \leq \frac{1}{m}\left(\sum_{i\notin\mathcal{I}_t}\|\mathbf{p}_{t,i}\| + \sum_{i\in\mathcal{I}_t}\|\mathbf{p}_{t,i}\|\right) \leq c_t\|\mathbf{g}_t\|,$$

where

$$c_t = \frac{1}{m\phi}\left(m + |\mathcal{I}_t| + \theta\sum_{i\in\mathcal{I}_t}\frac{1}{\delta_i}\right).$$

From this and an analogous argument to that in the proof of Theorem 1, for $\alpha \in (0, \omega_t]$, where

$$\omega_t = \min\left\{1, \frac{2(1-\rho)\theta}{Lc_t^2}\right\},$$

we have

$$\left\|\nabla f(\mathbf{w}_t + \alpha\mathbf{p}_t)\right\|^2 \leq \|\mathbf{g}_t\|^2 + 2\alpha\rho\langle\mathbf{p}_t, \mathbf{H}_t\mathbf{g}_t\rangle.$$

Therefore, line search (7) will pass for some $\alpha_t \geq \omega_t$ and $\|\mathbf{g}_{t+1}\|^2 \leq (1 - 2\omega_t\rho\theta)\|\mathbf{g}_t\|^2$. Upper bounding $c_t$ with

$$c = \frac{2}{\phi} + \frac{\theta}{m\phi}\sum_{i=1}^{m}\frac{1}{\delta_i},$$

which holds for all $t$, implies that line search (7) will pass for some $\alpha_t \geq \tau_3$, with

$$\tau_3 = \min\left\{1, \frac{2(1-\rho)\theta}{Lc^2}\right\} \leq \omega_t,$$

and $\|\mathbf{g}_{t+1}\|^2 \leq (1 - 2\omega_t\rho\theta)\|\mathbf{g}_t\|^2 \leq (1 - 2\tau_3\rho\theta)\|\mathbf{g}_t\|^2$. $\qquad\square$

## A.6   Lemma 3

*Proof of Lemma 3.* Suppose $|\mathcal{I}_t| < m$ for some iteration $t$. For such $t$ there exists an $i$ such that

$$\theta\|\mathbf{g}_t\|^2 \le \langle \tilde{\mathbf{H}}_{t,i}^\dagger \tilde{\mathbf{g}}_t, \mathbf{H}_t\mathbf{g}_t \rangle \le \|\mathbf{g}_t\| \cdot \left\|(\tilde{\mathbf{H}}_{t,i}^T)^\dagger \mathbf{H}_t\mathbf{g}_t\right\| \le \frac{1}{\phi}\|\mathbf{g}_t\| \cdot \|\mathbf{H}_t\mathbf{g}_t\|,$$

which implies $\theta\phi \le \|\mathbf{H}_t\mathbf{g}_t\|/\|\mathbf{g}_t\|$. $\qquad\qquad\square$

## B   Additional Experiments

In this section, we evaluate the empirical performance of DINGO, GIANT, DiSCO, InexactDANE, AIDE, Async-SGD and Sync-SGD on another softmax regression problem, a Gaussian mixture model problem and an autoencoder problem, which are convex, non-convex and non-convex, respectively. In our offline small-scale experiments we found that AIDE often failed to outperform InexactDANE in non-convex problems. Because of this and also the significant amount of difficulty in tuning its hyper-parameters, we opted not to consider AIDE in non-convex experiments.

We choose to run GIANT and DiSCO on non-convex problems, despite that their sub-problem solvers, conjugate gradient (CG) [24] and distributed PCG [9], are not meant to be used when the local Hessian matrices $\mathbf{H}_{t,i}$ and full Hessian $\mathbf{H}_t$, respectively, can be singular or indefinite. Moreover, we do not make any modifications on these sub-problem solvers, rather we terminate GIANT when CG fails and terminate DiSCO when distributed PCG fails. These situations are indicated by a cross "$\times$" in all plots. We make this choice to highlight situations where the iterates of GIANT and DiSCO enter areas that are non-convex or have a high degree of weak-convexity, and to highlight the difference, in convexity requirements, between DINGO and these methods.

### B.1   Softmax Regression

As in Section 4, we consider the strongly convex problem of softmax cross-entropy minimization with regularization. Here, we show the performance of the optimization methods applied to this problem on the EMNIST Digits dataset in Figure 3. This dataset has 240000 training samples, 40000 test samples and each datapoint $\mathbf{x}_i \in \mathbb{R}^{784}$ has a label $y_i \in \{1, 2, \ldots, 10\}$. This problem has dimension $d = 784 \cdot (10 - 1) = 7056$. EMNIST Digits has a large number of samples $n$ and we have $s = n/m > d$ in all experiments. As $s$ is relatively large, we see DiSCO perform comparatively poorly in Figure 3.

In Plot 3(d) we demonstrate the effect of choosing unnecessarily large values of $\theta$ for DINGO. In this experiment, each iteration of DINGO is in **Case 1** when $\theta = 10^{-4}$, **Case 1** and **Case 3** occur when $\theta = 1$, and each iteration is in **Case 3** when $\theta = 100$. We maintain a step size of 1 when $\theta = 1$ and $\theta = 100$ and we obtain similar convergence in the objective value for all three values of $\theta$. In Experiment 3(d), we obtain the exact same convergence for all values $\theta \le 10^{-1}$.

### B.2   Gaussian Mixture Model

For this experiment we consider learning the mixture weight and mean vectors of a mixture of two Gaussians. Namely, we have

$$\ell_j(\mathbf{w}) = -\log\left(\zeta(w_0)\Phi(\mathbf{x}_j; \mathbf{w}_1, \mathbf{\Sigma}_1) + \big(1 - \zeta(w_0)\big)\Phi(\mathbf{x}_j; \mathbf{w}_2, \mathbf{\Sigma}_2)\right) \tag{12}$$

where $\mathbf{w} = (w_0, \mathbf{w}_1, \mathbf{w}_2)^T$ and $\Phi$ denotes the density of the $p$-dimensional standard normal distribution. The data points $\mathbf{x}_i \in \mathbb{R}^p$ and covariance matrices $\mathbf{\Sigma}_1, \mathbf{\Sigma}_2 \in \mathbb{R}^{p \times p}$ are given. The function $\zeta : \mathbb{R} \to (0, 1)$ is defined by

$$\zeta(t) = \frac{1 + \tanh(t)}{2}.$$

This problem has dimension $d = 1 + 2p$. This problem is non-convex; however, it may exhibit features that are close to being invex [16].

The performance profile of each optimization method is presented in Figure 4 for various number of workers. In each experiment, we run each method a total of 100 times. Each time, we record the results of the iteration ending at, or immediately after, 20 communication rounds. Every run

|     |     |     |     |
| --- | --- | --- | --- |
| (a) 8 Workers | (b) 16 Workers | (c) 32 Workers | (d) 16 Workers |

Figure 3: Softmax regression problem on the EMNIST Digits dataset. All algorithms are initialized at $\mathbf{w}_0 = \mathbf{0}$. In Plots 3(a), 3(b) and 3(c): Async-SGD, Sync-SGD, SVRG have a learning rate of $10^{-2}$, $10^{-1}$ and $10^{-1}$, respectively, and AIDE has $\tau = 1$. In Plot 3(d) we compare the performance of DINGO for three different values of $\theta$, namely $10^{-4}$, 1 and 100. In this plot, similar convergence in the objective value is obtained, while DINGO with $\theta = 10^{-4}$ achieves a significantly faster reduction in the norm of the gradient.

we generate 20000 data points from the mixture distribution (12), with $p = 100$ and ground truth parameters $w_0^* = 1$, $\mathbf{w}_1^* \sim \mathcal{U}[-1, 0]$ and $\mathbf{w}_2^* \sim \mathcal{U}[0, 1]$. Moreover, every run we generate the covariance matrices $\boldsymbol{\Sigma}_1$ and $\boldsymbol{\Sigma}_2$ randomly in such a way that they are not axis-aligned and have a fixed condition number of 100, for details see [16]. In Figure 4 we also display the *estimation error*, see [16], which measures a model's relative accuracy in recovering the ground truth parameters $w_0^*$, $\mathbf{w}_1^*$ and $\mathbf{w}_2^*$.

## B.3 Autoencoder

For this experiment we consider a deep autoencoder on the Curves dataset. With this dataset we have $n = 20000$ training samples, 10000 test samples and each data point $\mathbf{x}_i$ is an element of $\mathbb{R}^{784}$. We use a fully-connected feed-forward autoencoder with bias, palindromic layer widths 784-400-300-200-100-50-25-12-6-12-...-784 and $\ell_2$ loss. On the hidden layers and output layer we apply the element-wise *sigmoid* and *exponential linear unit* (ELU) activation functions, respectively. The vector $\mathbf{w}$ exclusively contains the elements, in a one-to-one correspondence, of the sixteen weight matrices and sixteen bias vectors. This problem has dimension $d = 1043408$ and is non-convex [1].

We show the performance of the optimization methods applied to this problem in Figure 5. Note that the twice differentiable assumption, in our theory, is not supported in this problem. Nevertheless, DINGO empirically works and performs well. It is competitive in minimizing the objective value in all experiments in Figure 5. The hyper-parameters of DINGO remain constant among these experiments. Whereas, the hyper-parameters of InexactDANE, Async-SGD and Sync-SGD were selected carefully for them to achieve high performance, which is a time consuming process that is

Figure 4: Performance profiles on the Gaussian mixture model problem. All algorithms are initialized at $\mathbf{w}_0 = \mathbf{0}$. In all plots, Async-SGD, Sync-SGD and SVRG have a learning rate of $10^{-2}$, $10^{-1}$ and $10^{-1}$, respectively.

problem specific. Note the significant difference in selected hyper-parameters between Plot 5(a) and the other plots in Figure 5.

This experiment highlights some important differences between DINGO, GIANT and DiSCO. In Plot 5(a), all iterations of DINGO were in **Case 1**. In this situation, the update directions of DINGO and GIANT appear very similar, in theory, while their behaviour is noticeably different. They both achieve almost identical performance in the objective value; however, DINGO maintains a constant step-size of $1$ and achieves continual progress in reducing the norm of the gradient.

In Plots 5(b), 5(c) and 5(d) both GIANT and DiSCO are immediately terminated, whereas DINGO uses iterations from multiple cases to achieve a fast reduction in both the objective value and gradient norm. Specifically, in Plots 5(b) and 5(c) the first few iterations of DINGO are in **Case 1** and all subsequent iterations are in **Case 2** (except for the two iterations in Plot 5(b) corresponding to the dips in step-size, which are **Case 1**). Whereas, in Plot 5(d), **Case 1** occurs until the dip in step-size, where during this dip all iterations are in **Case 3**, and once the step-size returns to a value of $1$ all subsequent iterations are in **Case 2**. Figure 5 demonstrates the versatility of DINGO and how its cases allow it to traverse various regions.

Figure 5: Autoencoder problem on the Curves dataset. In Plot 5(a): Async-SGD, Sync-SGD and SVRG have a learning rate of $1$, $10$ and $1$, respectively. In Plots 5(b), 5(c) and 5(d): Async-SGD, Sync-SGD and SVRG have a learning rate of $10^{-3}$, $10^{-2}$ and $10^{-2}$, respectively. For Plots 5(b), 5(c) and 5(d) we use the default random initialization of PyTorch. In these three plots, both GIANT and DiSCO fail, i.e., they terminate before completing their first iteration. Recall that this is marked with a cross.