[Reviews · NeurIPS 2019]

Reviewer 1



In this paper, the authors propose a distributed Newton method for gradient-norm optimization. The method does not impose any specific form on the underlying objective function. The authors present convergence analysis for the method and illustrate the performance of the method on a convex problem (in the main paper). Originality: The topic of the paper, in my opinion, is very interesting. The paper presents an efficient Newton method that is motivated via the optimization of the norm of the gradient. As a result, no assumptions are made on the objective function. This is a key differentiating feature of the method as compared to other such methods. Quality: The overall quality of the paper is very good. The motivation is clear, the theory is well thought out and presented, and the numerical results show the superiority of the method (on a convex problem). I state my minor comments/questions/concerns below. Clarity: The paper is well-written and motivated. Below I state a few minor issues and concerns, as well as a few suggestions to improve the clarity and presentation of the work. Significance: In my opinion such distributed Newton methods are of interest to both the optimization and machine learning communities. This paper attempts to alleviate some of the issues (communication vs computation balance, number of hyper-parameters and sensitivity to hyper-parameters) of existing works. I believe that this paper, after minor corrections and modifications, should be accepted for publication at NeurIPS. Issues/Questions/Comments/Suggestions: - “increasingly more frequently”: is this phrase not redundant? - “regardless of the selected hyper-parameters”: The theory shows that this is indeed the case for the method. Of course, the method was designed, in a certain sense, such that that claim would be true. Nevertheless, it could be a bit misleading. Although, strict reduction is guaranteed with any hyper-parameter setting, this reduction could be very small if the hyper-parameters are chosen poorly. The authors should comment about this in the manuscript. - Related Work and Contribution paragraph 1: the authors should more clearly state which disadvantages apply to each method. - Line 66: “derived by optimization of the”: missing word? - Deriving DINGO from he optimization of the gradient norm is interesting and has many advantages as stated by the authors (e.g., no restrictions on the functions). However, does this approach have any limitations/drawbacks? For example, convergence to a local maximum or saddle point? The authors should add a comment about this in the main paper. - The discussion about the hyper-parameters is interesting, and shows the strength of the proposed approach. I suggest that the authors present this information in a table. For each method, the authors could clearly show the hyper-parameters associated. Moreover, in the table the authors could clearly state the per iteration cost (in terms of communications) of each method. - The drawback of the current analysis of DINGO is that it requires exact solutions to the sub-problems. The authors clearly state this. What is the limitation? How would the analysis change to account for inexact solves? The authors should comment about this in the manuscript. - Per iteration cost of DINGO: the authors should discuss the per iteration cost of DINGO in the manuscript. Both in terms of communication and computation, and compare with existing methods. If I am not mistaken, in the worst case, DINGO requires 4 rounds of communications per iteration, plus the number of communications associated with satisfying the line search condition. - Line Search: Usually, the Armijo condition does not have the factor of 2. The authors should comment about this in the paper. - DINGO algorithm is complicated: Algorithm 1 is complicated (with the three cases). The authors may want to give a high level description of the method before the present the actual algorithm. - Effect of theta parameter: The theta parameter controls the search direction chosen by the algorithm. Essentially, it controls that the search direction is not orthogonal to the gradient of (4), and that it is a descent direction. The authors should comment about this in the paper and discuss the role of theta. - Assumptions 3-6: The authors should add to the discussion of these assumptions. Why they are realistic? Why they are necessary? - In this experiment presented in the main paper, the function is strongly convex, and thus all iterates fall into Case 1. The authors should discuss the effect of Case 2 and 3 iterates on the performance of the method. - Future Work: What do the authors mean with “more efficient methods”?

Reviewer 2



Originality. As far as I understood, the paper proposes an extension of [15] for the distributed setup. This requires careful analysis, but not so many new ideas. I think the comparison with https://ieeexplore.ieee.org/abstract/document/8675539 should be added. Quality. The numerical experiments seem to be convincing. I didn't check the proofs thoroughly, but the high-level intuition is not clear for me. As far as I see, the method uses the inverse of individual Hessians H_i,t, but not the inverse of the full Hessian H_t. I don't understand, why this works in the light of the fact that H_t^{-1} != \sum H_i,t^{-1}. Also, it is not clear for me why the considered three cases are mutually exclusive. Clarity. Here I have the same questions as in the quality part. It would be nice to provide more explanations. At the same time, I very much appreciate that the authors give some intuition and illustration for their quite complicated assumptions and their connection to the standard assumptions. Significance. I think the method can be useful in practice. Especially since it has smaller number of hyperparameters, wider range of applications including non-convex functions. ===========After rebuttal============ I can't say that the authors addressed all my questions from the review. Especially the ones in the quality part. But, I found some answers in the GIANT paper. So, I leave my score unchanged.

Reviewer 3



As is stated in question 1, I like the idea of not using the strong convexity assumption. My main concern is associated with the computational costs of computing $H_{t,i}^\dagger g$ and\or $\widetilde H_{t,i} g$. It seems to me that (section 4) iterative methods are used to compute these matrix-vector multiplications. However, the convergence analysis seems to require exact computation. The authors are also suggested to elaborate more on the communication cost associated with the line search step. Other comments: 1. The algorithm DINGO involves a few hyper-parameters. It would be good if the authors can discuss how these hyper-parameters are tuned so that the algorithm can achieve better performance. 2. I am not sure whether the step length \alpha_t can eventually be chosen as 1. 3. For the convergence of the algorithm, many assumptions are needed (e.g., Assumptions 1,2,3,5). I am not sure whether the example considered in Section 4 satisfies the assumptions or not. 4. In the implementation, the authors use iterative solvers without preconditioners. If the subproblems have bad conditioner numbers, I do not know if these iterative methods can obtain solutions with sufficient accuracy to guarantee the progress of the algorithm.

Reviewer 4



Strength/weakness/questions/suggestions: 1- The paper is well-written and it is also structured properly. 2- In Algorithm 1, it is needed to calculate the product of pseudo-inverse of $\hat{H}$ and some vectors $g_t$ and $\tilde{g}_t$. It can be costly. It would be more clear if the authors clarify more about it. 3- In equation (6), it is expensive to calculate $P_{t,i}$. There is an inverse calculation (which the required info can be calculated by a system of equations too, however expensive), and in each iteration of the algorithm, there is some expensive parts mentioned in the above and current points plus “Line Search”. 4- In Algorithm 1, there is no explanation how “Line Search” will be done. Is it done in distributed environment? Or just in master machine? Also, the “Line Search” mentioned in this paper is very expensive (full gradient calculation is needed in each step), and if this “Line Search” is done in master node, then it may happen some cases that the master node would be very busy, or equivalently, the algorithm would not scale well (according to Amdahl’s law). 5- Assumption 3 seems to be a strong assumption. Also, the assumption in Lemma 1, “the Hessian matrix $H_{t,i}$ is invertible”, is strong too. Is the latter assumption is based on strong convexity? The reviewer did not notice why this assumption should be true. 6- In this paper, in several parts it is mentioned “a novel communication efficient distributed second-order optimization method”, however, there is no analysis regarding the required number of communication rounds to reach the solution that shows its efficiency (similar to DiSCO algorithm). 7- The reviewer could not find any information about how DINGO is distributed in practice? Do the authors use GPU or CPU for their calculations? The reviewer was eager to see the code for the proposed algorithm, however, no code was available to check how DINGO is distributed. 8- In Figure 1, it is suggested to compare the results based on wall-clock time not just the communication rounds. In some cases, the expensive calculations may be done in master node, therefore, less communication rounds would be needed, however, the wall-clock time would be very high. Another reason is that, the number of communication rounds is somehow equivalent to the number of iterations (not a good measure though), and it is suggested to compare the results of distributed algorithms based on true measure, i.e., wall-clock time. 9- In Figure 1, rows 1,2 and 4, did the authors consider the cost of line search when x-axis is “Communication Rounds” for DINGO? 10- In Figure 1, what do the authors mean by “Worker”? The details should be provided. 11- It is clear from row 3 of Figure 1, that the “Line Search” mentioned in Algorithm 1 is expensive (many gradient evaluation is needed), and not efficient if the authors want to have well-scaled algorithm. ============= After Rebuttal ============= I read the rebuttal carefully, and the authors answered my main concerns. I am generally satisfied with the rebuttal, and thus increased my score.

[Author Response · NeurIPS 2019]

We are grateful to all the reviewers for taking the time, reading our paper, and providing many useful comments. We are
particularly humbled by the fact that all reviewers were unanimously supportive of our work. Before addressing each
reviewer's individual questions, we make some general remarks regarding comments that were shared by the reviewers.

**Hyper-parameters:** As reviewers noted, while DINGO converges for any choice of hyper-parameters, correctly
choosing them will result in better performance. We have discussed this in Sections 3-4 and have provided theoretical
guidelines in Lemmas 1 and 3. Sensitivity to $\theta$ and the consequential effect of Cases 2-3 is examined in Appendix B.1.
We will move this result to the main paper, as the importance of such results was highlighted by reviewers.

**Assumptions:** Our assumptions (1, 2, 3, 5) are generalizations of those typically found in the literature. For example, if
each $f_i$ is smooth and strongly convex, then all our assumptions would be satisfied. As another example, if each $f_i$ is an
under-determined least squares objective, hence not strongly convex, Assumption 3 is still satisfied. In fact, Assumption
3 is weaker than requiring the pseudoinverse of each $\mathbf{H}_{t,i}$ is bounded in spectral norm.

**Sub-problems:** As noted in the paper, our analysis is limited to the exact solutions of the least-squares sub-problems,
which can be done in $\mathcal{O}(d^3)$ time. This is the same for computing exact solutions to the sub-problems of GIANT. In
practice, however, such least squares sub-problems can be solved inexactly using very efficient iterative methods. We are
actively in the process of developing inexactness theory for our future work, and the main challenges lie in guaranteeing
boundedness of the approximate solutions as well as ensuring sufficient descent using the average direction.

**Line-search:** Analogous to GIANT, backtracking line-search is performed distributively, and requires only 2 communi-
cation rounds. Each worker, in parallel, computes the gradient with each step-size from some predetermined list of
$k$ step-sizes, e.g., $1, 2^{-1}, 2^{-2}, \dots, 2^{-k+1}$, and the driver node then aggregates and checks the line-search condition
at each step-size. Alternatively, backtracking line-search can be done sequentially in which checking each step-size
takes 2 communication rounds. The term $\tau$ in Corollary 1, which is a lower bound on the step-size under all cases,
determines the maximum communication cost needed during line-search. We will elaborate on this in the paper.

**Comparison to Related Work:** On Page 2 we briefly compare the advantages and disadvantages of related methods.
Then, we elaborate further on Page 3. We will add a table to the main paper that summarizes the discussion on pages 2
and 3 . Such aspects include: restrictions on data distribution and functional form of the objective, requirements on the
degree of convexity/non-convexity, hyper-parameters and communication rounds per iteration. We will also include the
number of communication rounds required to achieve a solution, and under what metric. For example, for DINGO to
achieve $\|\mathbf{g}_t\| \leq \varepsilon$ then, by Corollary 1, it requires $\mathcal{O}\big(\log(\varepsilon)/(\tau\rho\theta)\big)$ communication rounds.

Below, we address each reviewer's specific comments:

**Reviewer #1: (i)** Indeed, as DINGO is minimizing the norm of the gradient, it may converge to a local maximum or
saddle point in non-convex problems that are non-invex. We have mentioned this in Future Work on page 8. However,
we agree and fully appreciate the need to highlight this disadvantage earlier on, e.g., in Contributions. We will do so
in the revision. **(ii)** The factor 2, in Armijo condition, arises as we multiply both sides by 2 to remove the $1/2$ from
equation (4). **(iii)** We will present the high level description (Section 2) before presenting Algorithm 1.

**Reviewer #2: (i)** Thank you for the reference. Indeed, it is relevant and very interesting, and we will reference it.
However, it appears to be suited to decentralized settings; whereas, our focus is on centralized methods. We aim to
compare extensively with it and similar methods, such as CoCoA, in future work. **(ii)** Each worker node uses its local
Hessian information to transform the full-gradient. This is similar to the method GIANT. However, unlike GIANT, we
don't impose restrictive assumptions that guarantee suitable descent automatically. Rather, we use three cases that are
designed to ensure suitable descent in the norm of the gradient, despite $\mathbf{H}_t^\dagger \neq \sum \mathbf{H}_{t,i}^\dagger$. Please refer to Remark 1.

**Reviewer #3: (i)** As correctly pointed out, the ability to eventually converge using the step-size 1 is one of the most
important aspects of Newton-type methods. To properly study this, we require local convergence analysis, which we are
actively pursuing as an extension to our work. It seems like that this property is tightly intertwined with the number of
workers and data distribution. **(ii)** Indeed, the issue of preconditioning is essential to the performance of iterative solvers.
In future work, we aim to evaluate preconditioning ideas, proposed by DiSCO, in our context and evaluate convergence.

**Reviewer #4: (i)** Please note that Lemma 1 is not an assumption about DINGO, nor is it a statement about its overall
performance. Rather, it illustrates the fact that if we were to make stronger assumptions, such as those in line with
GIANT, e.g., if $\mathbf{H}_{t,i}$'s were to be invertible, then DINGO can be written in its simplest form, i.e., Case 1. Please see to
Remark 1 for further discussion. **(ii)** Our implementation of DINGO uses Python with PyTorch on top of MPI and
supports GPU and CPU. It can easily train an existing PyTorch Module and we will provide a link to the code in the
final revision. **(iii)** We agree about the need to perform wall-clock time analysis, and we will do so extensively in future
work as part of the extension of our current analysis to inexact sub-problem solutions. This is because such comparison,
especially in distributed settings, is highly implementation dependent. **(iv)** All communication rounds, including those
from line-search, were considered in Figure 1. **(v)** We will clarify what we mean by "worker" in Figure 1.

[Meta-Review · NeurIPS 2019]

Dear Author(s), with reviewers, we have discussed many aspects of your paper and agreed that the main contribution of the paper is that it presents a novel, efficient, distributed second order method for ERM problems. As compared to existing algorithms (some of which have been published at ICML and NeurIPS), the algorithm has less hyper-parameters (and is less sensitive to the hyper-parameters), can be applied to a wider range of problems, out-performs these methods (in some cases with a significant margin) in the experiments shown in the paper, and has a better communication-computation balance. In terms of the analysis: The analysis is novel in the technique used. The algorithm is designed such that the search directions fall into three different categories. This way, the authors consider 3 different cases and combine them to get the overall convergence guarantees. The first two cases are well thought out and motivated. The third case is a safe guard that is seldom used in practice (note, this case never occurs in the strongly convex case). On the negative side, the worst-case complexity results of this method (I believe) is worse that first-order method. But that is to be expected, and I strongly believe that such argument should not be used to downgrade and reject the method (remember LBFGS has worse convergence guarantees than GD, but is far superior in practice). Another interesting contribution is the use of a line search. Line searches are not usually used by the ML community and we hope that your paper can bring a discussion. Many ML researchers argue that it is too expensive. I would like to encourage you if you can highlight the contribution (mentioned above) a bit more in your final (camera-ready version) paper. The reviewers also pointed out some minor issues and please make sure to address them in your final camera-ready submission.